# Outcomes and outcome measures reported in clinical studies of therapeutic mammaplasty: a systematic review protocol

Alice Lee 🆔 , Richard M Kwasnicki, Daniel R Leff

Department of Surgery and Cancer, Imperial College London, London, UK

**Correspondence to**
Dr Alice Lee;
alice.lee13@imperial.ac.uk

## ABSTRACT

**Introduction** Therapeutic mammaplasty (TM) is an oncological procedure which combines tumour resection with breast reduction and mastopexy techniques. Previous systematic reviews have demonstrated oncological safety of TM, but poor and inconsistent reporting of quality-of-life, aesthetic and functional outcomes, often with non-validated measurement tools. Moreover, there is a paucity of patient-reported outcome measures. Standardisation of outcome reporting is required to enable study results to be compared and combined, for example, through core outcome set (COS) development. This systematic review aims to comprehensively describe the outcomes reported in clinical studies of TM, their respective outcome measures and the time points at which they were evaluated. The overall objective is to facilitate the development of a COS for TM.

**Methods and analysis** A systematic review of clinical studies evaluating outcomes following TM will be completed according to the Preferred Reporting Items for Systematic Reviews and Meta-Analyses (PRISMA) guidelines. The following electronic databases have been searched from inception to 5 August 2020: Ovid MEDLINE, Embase, CINAHL and Web of Science. Primary outcomes will include the number of reported outcomes of various types (clinical, aesthetic, functional, quality-of-life and cost-effectiveness), whether these are patient-reported or clinician-reported, how outcomes are defined and the outcome measurement tool(s) used. The time point(s) at which outcomes were measured will be a secondary outcome. No studies will be excluded on the basis of methodological quality in order to generate a comprehensive list of reported outcomes and outcome measures; hence, risk of bias assessment is not required. The data will be described narratively. This protocol has been reported in line with PRISMA-Protocols.

**Ethics and dissemination** This study does not involve human or animal participants, hence ethical approval is not required. The findings will be published in a peer-reviewed journal and presented at relevant conferences.

**PROSPERO registration number** CRD42020200365.

## INTRODUCTION
### Background
Therapeutic mammaplasty (TM) is an oncological procedure which combines cancer resection with breast reduction and

### Strengths and limitations of this study

► This systematic review will facilitate the development of the first core outcome set (COS) for therapeutic mammaplasty (TM).
► A COS will improve the synthesis and meta-analysis of new and existing research into TM.
► The systematic review described in this protocol will build on existing literature by providing an updated review of outcomes following TM with a broader scope, including clinical, aesthetic, functional, quality-of-life and patient-reported outcomes.
► The literature search was limited to English language articles, which may have excluded some relevant papers.

mastopexy techniques.[1] TM can facilitate breast-conserving surgery (BCS) for larger tumours[2] to safely avoid mastectomy[3] and improve cosmesis in cases where standard BCS would otherwise have poor outcomes.[4] TM may also minimise radiotherapy-related side effects in women with larger breasts[4 5] and be a favourable option for women with pre-existing macromastia who seek the functional and psychological benefits of breast reduction surgery.[4] Previous systematic reviews have suggested satisfactory oncological safety of TM,[2 4 6] but poor and inconsistent reporting of quality-of-life, aesthetic and functional outcomes, often with non-validated measurement tools.[5 7 8] Moreover, there is a paucity of patient-reported outcome measures.[6]

A core outcome set (COS) describes the minimum number of outcomes to be reported across all trials of one healthcare domain.[9] This reduces the heterogeneity of outcome reporting across trials, allowing results to be compared and combined in meta-analyses, to inform best medical practice.[9] TM is becoming routine practice in oncoplastic breast units, however there is no standardised way to evaluate outcomes

following this procedure which incorporates the views of both healthcare professionals and patients as stakeholders. A related COS on reconstructive breast surgery[10] mainly focused on post-mastectomy reconstruction; only 10% of patient stakeholders in the project had undergone TM and some outcomes included in the final COS (eg, implant-related complications) are less relevant to the TM population. There is good reason to hypothesise that patients who had TM may evaluate and prioritise their treatment outcomes differently to patients undergoing other forms of breast reconstruction. For example, improved functional outcomes associated with breast reduction techniques and avoidance of mastectomy may significantly drive treatment decisions.

### Rationale

In order to develop a COS for TM, a comprehensive systematic review of all available outcomes and outcome measures reported in the literature is required. The systematic review described in this protocol is the first stage in the development of a COS for TM, which is planned and has been prospectively registered on the Core Outcome Measures in Effectiveness Trials (COMET) database (http://comet-initiative.org/Studies/Details/1655).

### Aims and objectives
#### Review aims

The overall aim of this systematic review is to identify all outcomes and outcome measures used to evaluate TM in the literature and how the authors define these outcomes. The time points at which these outcomes are measured will be a secondary outcome.

#### Objectives

The specific objectives of this review are to analyse all clinical studies of TM in adult, female participants in order to:

i.   Identify the number of unique outcomes and outcome measures reported;
ii.  Identify and describe variation in outcome definitions;
iii. Identify and describe variation in the time point(s) used to measure outcomes;
iv.  Identify the number of different types of outcome (clinical, aesthetic, functional, quality-of-life and cost-effectiveness outcomes) reported per study and across all included studies and whether these are clinician- or patient-reported;
v.   Group unique outcomes into domains, to facilitate the development of a COS for TM.

### METHODS AND ANALYSIS

This protocol has been developed in accordance with the Preferred Reporting Items for Systematic Reviews and Meta-Analysis Protocols (PRISMA-P) guidelines.[11] A PRISMA-P checklist for this protocol can be found in online supplemental material 1. The systematic review described in this protocol has been prospectively registered on PROSPERO. Any amendments to this protocol, involving screening, data extraction or data synthesis will be documented and referenced in any subsequent, related publications.

### Eligibility criteria

This systematic review will include clinical studies of adult, female participants who have undergone TM as primary treatment for breast cancer. For the purpose of this systematic review, TM will be defined as the use of oncoplastic reduction or mastopexy techniques, including removal of the skin envelope and/or nipple if indicated, to treat pre-invasive or invasive breast cancer with BCS.[12] The eligibility criteria are summarised in table 1. At the time of writing this protocol, screening (by title and abstract) has commenced. The initial search returned 5709 de-duplicated articles.

**Table 1** Inclusion and exclusion criteria for the systematic review

| Inclusion criteria | Exclusion criteria |
|---|---|
| ► Randomised and non-randomised trials, cohort studies and case–control studies.<br>► Adult female participants undergoing TM as primary treatment for breast cancer (including both immediate and delayed symmetrisation).<br>► TM techniques (level 1–2 oncoplastic breast surgery) including the following skin incision patterns: wise, vertical scar, periareolar or circumareolar, Grisotti, melon slice (horizontal wedge excision). | ► Wrong study design: systematic reviews, meta-analyses, case series, case reports, conference abstracts and animal, cadaveric or laboratory studies.<br>► Non-English language articles.<br>► Non-oncological breast surgery.<br>► Studies which do not report TM techniques (including total mastectomy with or without reconstruction/ symmetrisation or standard BCS).<br>► Studies with male participants or those who are <18 years old.<br>► Articles which do not report patient outcomes.<br>► BCS combined with volume replacement procedures including but not limited to implants, latissimus dorsi mini-flaps, thoracodorsal artery perforator flaps, lateral intercostal artery perforator flaps. |

BCS, breast conserving surgery; TM, therapeutic mammaplasty.

**Table 2** Example search strategy for Ovid MEDLINE

| Search concept | | |
| --- | --- | --- |
| **Therapeutic mammaplasty** | **Breast cancer** | **Study design** |
| (1) (therapeutic adj3 mamm??plast*).mp. | (6) exp Breast Neoplasms/ | (12) randomized controlled trial.pt. |
| (2) reduction mamm?plast*.mp. | (7) breast neoplasm*.mp. | (13) controlled clinical trial.pt. |
| (3) oncoplastic breast surg*.mp. | (8) (breast adj2 cancer*).mp. | (14) randomi?ed.ab. |
| (4) Mammaplasty/ | (9) (breast adj2 tumo?r*).mp. | (15) placebo.ab. |
| (5) 1 or 2 or 3 or 4 | (10) 6 or 7 or 8 or 9 | (16) drug therapy.fs. |
| | | (17) randomly.ab. |
| | | (18) trial.ab. |
| | | (19) groups.ab. |
| | | (20) 12 or 13 or 14 or 15 or 16 or 17 or 18 or 19 |
| | | (21) exp cohort studies/ |
| | | (22) cohort$.tw. |
| | | (23) controlled clinical trial.pt. |
| | | (24) epidemiologic methods/ |
| | | (25) limit 24 to yr="1966–1989" |
| | | (26) exp case-control studies/ |
| | | (27) (case$ and control$).tw. |
| | | (28) 21 or 22 or 23 |
| | | (29) 24 or 25 or 26 or 27 |
| | | (30) 28 or 29 |
| (11) Combined search for therapeutic mammaplasty AND breast cancer = (5 AND 10) | | (31) Trial OR cohort study OR case-control study= (20 OR 30) |
| Overall search = (11 AND 31) | | |

## Information sources

The following electronic databases have been searched from inception to 5 August 2020: Ovid MEDLINE, Embase, CINAHL and Web of Science. The reference lists of included articles will also be hand-searched. The outcomes generated from this review will be cross-referenced with those reported in the Oncoplastic Breast Reconstruction Guidelines for Best Practice co-produced by the Association of Breast Surgery and British Association of Plastic Reconstructive and Aesthetic Surgeons.[13] Outcomes from these national documents which are relevant to the TM population and not already included in the review will be added.

## Search strategy

An example search strategy for Ovid MEDLINE is provided in table 2. The search strategies for Embase, CINAHL and Web of Science can be found in online supplemental material 2. In order to focus the search and make screening numbers manageable, validated study design filters for clinical trials, cohort studies and case–control studies were used.[14 15]

## Study records

Covidence software (V.2103) will be used for de-duplication of citations and article screening. Two researchers will independently screen papers against inclusion criteria in two stages: (i) by title and abstract, and ii) by full-text. Data will be extracted using a piloted form in Microsoft Excel (V.16.41), developed for the purposes of this review. Included studies will be extracted independently by two researchers. The first 10 included full-texts will be extracted to trial the suitability of the data extraction sheet. Any conflicts which arise in the screening or extraction stages will be resolved through discussion between the relevant researchers; if they cannot be resolved a third researcher (RMK or DRL) will make the final decision.

## Data items

The following variables will be extracted: study details (publication year, study design, TM procedure, average follow-up time) and information on study population (n number and average age). Outcome data will include total number of reported outcomes, whether outcomes are clinician-reported or patient-reported, how study authors have defined each outcome, the outcome measure(s) used and the time point(s) at which each outcome is measured. Outcomes will be categorised (into clinical, aesthetic, functional, quality-of-life and cost-effectiveness outcomes) by the researchers performing data extraction. Clinical outcomes will include oncological outcomes and operative complications. Aesthetic outcomes will include all measures of satisfaction with postoperative appearance, either clinician-reported or patient-reported. Functional outcomes will include, but are not limited to, level of physical activity, neck, shoulder, breast or back pain and intertrigo. Quality-of-life outcomes will include domains such as psychosocial, sexual and physical well-being. Cost-effectiveness outcomes will include formal analysis of cost-effectiveness or surrogate measures such as length of stay. Outcome measures which fall into more than one of the aforementioned categories will be reported as such and described narratively.

## Outcomes and prioritisation

The primary outcomes of this study will include the number of unique outcomes and outcome measures reported in studies of TM and any reported variation in outcome definition between studies. As a secondary outcome, we will summarise the time points used for outcome measurement and any variation between studies with respect to the timing of outcome measurement.

## Risk of bias in individual studies

No studies will be excluded on the basis of methodological quality in order to generate a comprehensive list of reported outcomes and outcome measures; hence, risk of bias assessment is not required.

## Data synthesis

The following data will be summarised quantitively: number of outcomes categorised as clinical, aesthetic, quality-of-life and cost-effectiveness per study; the number of different definitions used for each outcome across all included studies and the number of different outcome measures and time points used to measure each outcome across all included studies. The percentage of included studies which evaluate aesthetic, functional, quality-of-life, cost-effectiveness and patient-reported outcomes will be calculated. Outcomes will then be grouped into domains using an existing[9] or author-generated ontological framework, depending on the final list of outcomes obtained.

## Patient and public involvement

Patients were not directly involved in the design or conduct of this systematic review, since no participant recruitment will take place and the research is based on previously published data.

## ETHICS AND DISSEMINATION

This study does not involve human or animal participants, hence ethical approval is not required. The completed systematic review will be presented at relevant academic conferences, reported in accordance with PRISMA guidelines[16] and published in a peer-reviewed journal.

**Acknowledgements** The authors gratefully acknowledge Mr Michael Gainsford (St Mary's Hospital Library, Imperial College London) for his support developing the literature searches for this systematic review.

**Contributors** AL generated the systematic review question, design and search terms. AL drafted the protocol manuscript. DRL and RMK critically reviewed the systematic review question and design and reviewed this protocol manuscript.

**Funding** This work is independent research funded by the National Institute for Health Research (NIHR) Imperial Biomedical Research Centre (BRC). The views expressed in this publication are those of the authors and not necessarily those of the NHS, the NIHR or the Department of Health.

**Competing interests** None declared.

**Patient consent for publication** Not required.

**Provenance and peer review** Not commissioned; externally peer reviewed.

**ORCID iD**
Alice Lee http://orcid.org/0000-0002-8492-837X

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
