## [Reviewer comments · BMJ Open]

ARTICLE DETAILS

TITLE (PROVISIONAL)	Outcomes and outcome measures reported in clinical studies of therapeutic mammoplasty: A systematic review protocol
AUTHORS	Lee, Alice; Kwasnicki, Richard; Leff, Daniel

VERSION 1 – REVIEW

REVIEWER	Rose, Michael Sydvestjysk Sygehus Esbjerg, Plastic Surgery
REVIEW RETURNED	13-Dec-2020

GENERAL COMMENTS	No comments
-------------

REVIEWER	Di Giacomo, Dina Università degli Studi dell'Aquila, Department of Life, Health and Environmental Sciences
REVIEW RETURNED	24-Jan-2021

GENERAL COMMENTS	The Proposed Study Protocol is very interesting and becoming an emergent topic in cancer surgery treatments as well aesthetic surgery. However, I suggest to report the range time of reviewing studied involved in the study analysis; more, a better description of
--

REVIEWER	Karakatsanis, Andreas Uppsala Universitet
REVIEW RETURNED	06-Mar-2021

GENERAL COMMENTS	The review intended is highly relevant and appropriate. Standard setting and definition of COS is an imperative priority in clinical research regarding oncoplastic breast surgery, as current research is largely opinionated. The protocol is appropriate but refinement may allow for improvement. Authors intend to follow the PRISMA guidelines and consider surgery as intervention but this is not optimal as there is a lack of trials with the intention of one vs another surgical intervention; most of the data expected stems from source articles where surgery was decided and the outcome reported afterwards. This is by definition observational and the MOOSE guidelines are more appropriate. Moreover, authors intend a narrative presentation of the outcomes; albeit challenging, this may allow for a better critical appraisal of source studies and definition of knowledge gaps from the available literature in a structured manner. The latter is particularly important as knowledge gaps have insofar been addressed only in the context of expert panel opinions and modified Delhi consensus meetings, which is still expert opinion and, thus, a low level of evidence. In this context,
--

	perhaps the authors would like to consider an initial step of a scoping review to the lead to a systematic review, as this is the most elegant methodology.
--	---

VERSION 1 – AUTHOR RESPONSE

Reviewer: 1

Dr. Michael Rose, Sydvestjysk Sygehus Esbjerg

Comments to the Author: No comments

This reviewer did not provide any comments.

Reviewer: 2 Dr. Dina Di Giacomo, Università degli Studi dell'Aquila

Comments to the Author:

The Proposed Study Protocol is very interesting and becoming an emergent topic in cancer surgery treatments as well aesthetic surgery.

We thank this reviewer for highlighting the importance of our proposed review in the field of oncological and aesthetic surgery.

However, I suggest to report the range time of reviewing studied involved in the study analysis; more, a better description of

*****Comment from the Editor: We are sorry that reviewer 2's comments were incomplete. We were unable to confirm with them how this sentence should end.*****

We understand that Dr Giacomo's comments were unfortunately incomplete. However, we appreciate that she raised a point regarding the time range of studies included in the analysis. We

can reassure Editors and reviewers that there is no time limit on our database search, so we will comprehensively cover all clinical studies published in the field to date. This is reflected in Tables 1 and 2, which demonstrate that publication year was not an exclusion criterion or search filter. Secondly, we will extract and report data on year of publication and average study follow-up time (described under 'Data items' on page 8), to allow readers to contextualise the timing of outcome reporting.

Reviewer: 3 Dr. Andreas Karakatsanis, Uppsala Universitet

Comments to the Author:

The review intended is highly relevant and appropriate. Standard setting and definition of COS is an imperative priority in clinical research regarding oncoplastic breast surgery, as current research is largely opinionated.

We thank this reviewer for highlighting the importance of our proposed review in the field of outcome reporting in oncoplastic breast surgery.

The protocol is appropriate but refinement may allow for improvement. Authors intend to follow the PRISMA guidelines and consider surgery as intervention but this is not optimal as there is a lack of trials with the intention of one vs another surgical intervention; most of the data expected stems from source articles where surgery was decided and the outcome reported afterwards. This is by definition observational and the MOOSE guidelines are more appropriate.

We appreciate the important points raised by this reviewer and would like to clarify them. Since we are conducting a systematic review, we will report this in line with PRISMA guidelines (not MOOSE guidelines), as per best practice. We will include all articles which report the intervention of interest, i.e., therapeutic mammoplasty. We appreciate that the majority of the evidence base will be single cohort studies of therapeutic mammoplasty and not randomised-controlled trials or trials with

comparators. We would like to reassure the Editors/reviewers that these observational studies will definitely be included (*Table 1*) and will make up the bulk of included articles. We have chosen to assess outcome reporting of observational studies using the methodology described, as opposed to comparing with MOOSE guidelines. Our methodology is in line with other Core Outcome Set development projects and the COMET (Core Outcome Measures in Effectiveness Trials) Handbook.

Moreover, authors intend a narrative presentation of the outcomes; albeit challenging, this may allow for a better critical appraisal of source studies and definition of knowledge gaps from the available literature in a structured manner. The latter is particularly important as knowledge gaps have insofar been addressed only in the context of expert panel opinions and modified Delhi consensus meetings, which is still expert opinion and, thus, a low level of evidence.

We agree with the reviewer that a narrative description of outcomes and deficiencies in outcome reporting is most appropriate, and that our review will contribute high level evidence to this topic.

In this context, perhaps the authors would like to consider an initial step of a scoping review to the lead to a systematic review, as this is the most elegant methodology.

We agree with the reviewer that this review is ambitious and covers outcome reporting in a broad sense. We thank the reviewer for putting forward the idea of a scoping review. We agree that obtaining a general sense of outcome reporting and the outcome measures used is useful before embarking on a systematic review of this kind. As a research team, we did this by searching the literature which we used to inform our protocol. We believe that publishing a separate scoping review may duplicate the findings of our systematic review. Furthermore, the rationale behind this project is to provide a comprehensive overview of outcome reporting in therapeutic mammoplasty. This can only be achieved using systematic methods.

Editorial office

Kindly embed your Patient and Public Involvement statement under methods section of your main document.

This has been amended.

Please re-upload your supplementary files in PDF format. - Supplementary material_PRISMA-P-checklist.doc (v1.0)

This has been amended.

Please change the label of your Supplementary file and cite it as "Supplementary file 1", "Supplementary file 2" to avoid confusion. Please also ensure to cite it all in the main text of your

main document file.

This has been amended.

Reference citations should be cited in ascending order. You have cited 'reference 12' right after 'reference 14' which makes your citations incorrect. Please review again the main document and ensure that all references are cited in ascending order.

This has been amended.

Other changes made by the authors

We would like to draw your attention to three other minor changes which have been made.

- In '*Information sources*' (line 145) we mentioned cross-referencing reported outcomes with two national audits in breast surgery. The first of these (National Mastectomy and Breast Reconstruction Audit) has been removed, because it pertains to mastectomy which is irrelevant to the therapeutic mammoplasty population. The other '*Oncoplastic breast reconstruction guidelines*' has been retained.
- Under '*data synthesis*' (line 197), we have removed the line which states we will report the number of outcomes reported per study. We believe it is more clinically and scientifically meaningful to describe what has been reported, and that numbers without context will not add much.
- Exclusion and inclusion criteria (Table 1):
 - o We have changed one of the exclusion criteria to <18 years as opposed to <16 years, which is a more usual age cut-off for adults.
 - o We have clarified the definition of TM (level 1-2 oncoplastic breast surgery) and we have added more examples of excluded volume replacement procedures as clarification for the reader.

We hope that the above responses to the reviewer comments are satisfactory. We would be very happy to discuss any ongoing concerns and modify our manuscript as required.

VERSION 2 – REVIEW

REVIEWER	Karakatsanis, Andreas Uppsala Universitet
REVIEW RETURNED	28-Apr-2021

GENERAL COMMENTS	Whilst I understand the rationale behind not choosing MOOSE guidelines, the approach the authors choose is suboptimal in terms of methodology. However, the results may not differ substantially, as the "intervention" is not an intervention and no meta-analysis is intended. Additionally, the suggestion regarding a scoping review did not imply a second publication; instead, the point is that, if performed first, it would allow for refinement of the systematic review endpoints. I think that the cause is nobler than the details, and I trust authors will assess any pitfalls in data extraction and interpretation and synthesis of findings.
--